# Impact of Local Anesthetics on Cancer Behavior and Outcome during the Perioperative Period: A Review

**DOI:** 10.3390/medicina58070882

**Published:** 2022-06-30

**Authors:** Alain Borgeat, José Aguirre

**Affiliations:** 1Balgrist Campus, University of Zurich, Lengghalde 5, 8008 Zurich, Switzerland; josealejandro.aguirre@stadtspital.ch; 2Institute of Anesthesiology, Triemli City Hospital, 8063 Zurich, Switzerland

**Keywords:** local anesthetics, metastases, cancer recurrence

## Abstract

There is a growing interest regarding the impact of the perioperative period and the application of anesthetic drugs on the recurrence of cancer metastases. Among them, the use of amide-type local anesthetics seems promising since in vitro studies and animal models have shown their potential to inhibit the Intercellular Adhesion Molecule 1 (ICAM-1) expression and Src activity, which are clearly implicated in the process of inflammation and cancer metastases. This review emphasizes the potential of amide-type local anesthetics in this context.

## 1. Introduction

Lidocaine is an old drug, which has been acknowledged to be exclusively indicated for the management of pain and cardiac arrhythmias through its ability to block the sodium channel; however, new research has demonstrated that this drug was able to block or interfere with other receptors and systems leading to new clinical development. One of them could be the potential to reduce cancer metastases.

Cancer is still one of the major causes of morbidity and mortality worldwide [1]. Surgical removal is still used in more than 60% of cases [2]; however, the perioperative period is increasingly recognized as a time point with the release of circulating tumor cells and a great influence on the immune system [3].

Recently more and more evidence has shown that inflammation and metastases are closely connected [4]. The presence of leucocytes within and around tumors was already observed by Virchow in the 19th century leading to the evidence that an inflammatory microenvironment was an essential component of all tumors [5].

Metastases are the most crucial aspect of tumorigenesis because approximately 2/3 of cancer mortality is caused by metastases [6,7]. This process requires a close link between cancer cells and immune and inflammatory cells. Among the different steps necessary for the occurrence of metastases, one involves direct inflammation, which, through the release of mediators, increases vascular permeability. In this context, it was shown that local anesthetics (LAs) possess anti-inflammatory properties [8,9] by acting, among others, on the G protein-coupled receptors [10].

Surgery is a known model to favor the development of cancer: There is a great inflammatory response, and the immune defenses are greatly impaired. Lin et al. [11] have extensively described surgical-induced systemic inflammation. Primarily, this process should eradicate microorganisms and enhance healing; however, this response can be exaggerated, which will impair the immune system, leading to multiple organ failure and patient death [12]. This occurrence may have the potential to favor metastases.

## 2. Circulating Tumor Cells (CTCs)

The role of CTCs has been hypothesized to play a major role in the formation of metastases [13]. Recently CTCs have been studied as a biomarker for cancer detection and prognosis. It was shown that the number of CTCs was associated with patient survival in different types of cancer, including breast, colon, and many others [14]. Asworth [15] was the first to describe the occurrence of CTCs, but the clinical interest in this finding was postponed until recently due to technical issues for isolating the CTCs since they are ultra-rare events. Moreover, they are derived from different types of tissues, knowing that each one distinguishes from the other considering the size, shape, and immune phenotyping profile [16].

The CTCs are considered to be shed from different locations within tumors; however, this occurrence is accelerated during surgery. Sergeant et al. [17] have found that after a successful pancreatectomy for an adenocarcinoma, the outcome was poor due to new distant metastases related to a large increase in CTCs [17,18]; however, the diameter of CTCs is large, greater than the bores of capillaries, in which we would expect that these cells would be blocked [19]. This concept suggests that other mechanisms should be activated to allow the CTCs to transfer directly into the tissues from the vascular endothelia. To allow this occurrence, some biomolecular changes should occur within the CTCs, making them more aggressive. This can occur secondary to the severe inflammatory reaction induced by the surgery.

## 3. Anti-Inflammatory Effects of Local Anesthetics Potentially Affecting Metastases Immune Modulation/Natural Killer (NK) Cell Activity

Each surgery or trauma will create new body homeostasis, called the surgical-induced stress response [20,21]. This phenomenon is characterized by the production of catecholamine and the activation of the corticotropic axis leading to a redistribution of circulating leucocytes, resulting in lymphopenia [22]; therefore, during this period, a generalized pro-inflammatory state occurs. It is associated with the release of different cytokines, including, among others, IL-6 and tumor necrosis factor, [23] which can potentially favor tumor progression. This inflammatory surrounding also negatively influences the immune system [24].

In humans, NK cells have been the most studied. NK cells were first described by Kiessling et al., 1975 [25], and it became rapidly evident that these cells play an important role in establishing anti-tumor immunity [26]. In humans, Iannone et al. found that NK cell cytotoxicity was significantly inhibited in pancreatic cancer patients after pancreatectomy. It was noted that only after 30 days the NK cell activity was restored [27]. A reduction in NK cell cytotoxicity was also observed after surgery for pulmonary, hepatocellular, and breast cancer [28]. In animals in whom tumor cells were administered IV, tumor cell retention was significantly increased after laparotomy compared to those having sham surgery [29].

The interaction of different anesthetics on the NK cell activity has been tested. Among the volatiles, halothane has been shown in rats to reduce the cytotoxicity of NK cells [30]. In the same investigation, retention of tumor cells was increased with halothane, but not after administration of propofol [30]. Among analgesics, morphine has been shown in animals and humans to have a negative effect on NK cell activity (Table 1).

A dual effect of lidocaine has been observed. At a very high concentration in vitro, lidocaine has been shown to have a negative effect [31], whereas, at clinical concentration, lidocaine demonstrated an increase in the cytotoxic potential of NK cells [32]. The presumed beneficial action was due to the release of granzyme b and perforin [32]. Moreover, the serum from patients receiving lidocaine during cancer surgery was greatly competent in killing cancer cells [33,34]. Another beneficial effect of lidocaine is the action of the Th1/Th2 ratio. CD4 and T-cells, after antigen stimulation, will differentiate into Th1 or Th2 cells. Interferon-γ (IFN-γ) is secreted by Th1 cells and is responsible for cellular immunity. On the other hand, Th2 cells are associated with humoral immunity through the release of IL-4. The proportion of these mediators determines the Th1/Th2 ratio. Knowing that cellular immunity is a key factor in controlling the immune response toward the tumor, any reduction in this ratio will favor tumor progression. In patients undergoing hysterectomy or hepatectomy, the addition of lidocaine inclined the ratio Th1/Th2 towards a Th1 profile, including the production of IFN-γ [35,36]. Numerous studies have demonstrated the negative effect of certain cytokines, especially IL-6, which has been associated with tumor progression [37,38]. These mediators increase dramatically after all surgeries and/or trauma. Investigations have shown that the perioperative application of lidocaine significantly decreased the release of pro-inflammatory cytokines [39]. This suggests an intensive cross-talking between inflammatory and tumor cells, making a common pathway between these two systems likely and emphasizing the interest in giving lidocaine during this period to stimulate the immune system and protect against tumor cell dissemination during the perioperative and first postoperative days.

## 4. Endothelial Barrier, Leucocyte Activation, Leukocyte/Tumor Cell Adhesion, and Transmigration

The prerequisites for metastatic dissemination of solid tumors are migration, invasion, and adhesion of cancer cells. The movement mechanism of the cells is migration, while direct extension and penetration of cancer cells into tissues is invasion. In between, the process of adhesion also plays a critical aspect in the formation of metastases. It seems that this process is involved in the great majority of metastases. Amide-type LAs have been shown to interact positively with each of these steps. In a model of endotoxin-induced lung injury in rats, the application of ropivacaine IV was shown to significantly decrease ICAM-1 expression, as well as neutrophil adhesion and the concentration of albumin in bronchoalveolar lavage [40]. Piegeler et al. [41] exposed mice to either nebulized normal saline or lipopolysaccharide. The addition of ropivacaine showed a significant reduction in excess lung water, permeability index, and myeloperoxidase activity compared to the control. In the treatment group significant reduction in Src activation/expression, as well as ICAM-1 expression and caveolin-1 phosphorylation, was observed. It was concluded that ropivacaine was efficient in this model to treat the cause of pulmonary edema. Another study demonstrated that lidocaine and ropivacaine blocked inflammatory TNFα signaling in pulmonary endothelial cells by attenuating p85 recruitment to TNF-receptor 1, resulting in decreasing Akt, endothelial nitric oxide synthase, and Src phosphorylation [42]. Src activation has been shown to be responsible for a massive loss of endothelial barrier function, [43] which may also contribute to an increase in the extravasation of CTCs [44]. ICAM-1 is expressed by many different types of cancer cells [45,46,47,48] and may play an important role in the adhesion of CTCs on the endothelium, resulting in enhancing the extravasation of CTCs [47,48,49,50,51]. Other investigations have also shown that polymorphonuclear neutrophils (PNMs) activation and priming were significantly reduced by LAs [52,53,54,55]. If most of the studies pointed out beneficial effects, some did not show any benefit, which could be explained by methodological issues [56].

There is some evidence in vitro and animal models that LAs in cancer patients undergoing surgery may have the potential to reduce CTCs extravasation during the perioperative period by preserving the endothelial barrier function, reducing the PMNs adhesion, and therefore having the potential to reduce the transmigration of CTCs.

## 5. Direct Effects of LAs on Cancer Cells

LAs may have the potential to reduce/inhibit the occurrence of metastases by targeting different receptors/mechanisms (Table 2). Piegeler et al. [57] investigated in vitro the effects of lidocaine and ropivacaine on human lung cancer cells in the presence or not of an inflammatory stimulus (TNFα). The authors found that at clinical concentrations, both drugs in both conditions inhibited the Src expression and the ICAM-1 expression. The migration of cancer cells through a biological membrane was significantly reduced when the cells were in contact with the drugs for 4 h, but not if the exposure was limited to 15 min, followed by a washout. In these experiments, it was shown that these effects were not influenced by the addition of veratridine or tetrodotoxine, suggesting that these observations were independent of the sodium channel function. Moreover, chloroprocaine, an ester-type LA, did not show any effect. This work suggested that the positive effects of lidocaine and ropivacaine are time- and concentration-dependent, are not linked to any interaction with the sodium channel and are specific for the amide-type of LAs. Local anesthetics have also been shown to interfere with the regulation of gene expression by changing the methylation of DNA of breast cancer cells in vitro. In this investigation, lidocaine was able to demethylate DNA of these cells, a mechanism that will help to eliminate the cancer cells. Another investigation demonstrated that colon cancer cells’ invasiveness was decreased in vitro by the sodium channel 1.5 blocking effect [58]. Koh et al. [59] investigated the effect of lidocaine added to cisplatin in patients having highly aggressive triple-negative breast cancer. They found that this combination significantly increased the apoptosis ratio and the inhibitory effects of cisplatin. A higher expression of activated caspase-3 was observed. Lidocaine is also considered an ion channel regulator, which can regulate channel or membrane potential, such as mitochondrial membrane potential resulting in the mitochondria-related apoptosis [60]. Du and Coll [61] investigated in vitro the effects of lidocaine on the proliferation of colorectal cancer cells. The authors found that the proliferation of these cells was suppressed by the blockade of aerobic glycolysis.

## 6. Human Retrospective Studies

The interest in lidocaine application in cancer patients has been the subject of many clinical studies in recent years.

A retrospective analysis of medical records of a patient who had a mastectomy and axillary dissection for breast cancer was performed by Exadaktylos et al. [67]. Two groups were compared; one received general anesthesia (GA) alone, in the other, a paravertebral block was added. The results showed that the paravertebral group had at 37 months a better metastasis/recurrence-free survival of 94% compared to 77% in the control group.

In a study including 2239 patients undergoing pancreatectomy for adenocarcinoma, Zhang et al. [68] found that the addition of a bolus of lidocaine followed by a continuous infusion until the end of the surgery was associated with prolonged overall survival but without prolonged disease-free survival. The major weakness of this investigation was the short time of lidocaine administration, knowing that in vitro studies have shown that the positive effect of lidocaine in this context is time- and concentration-dependent.

Another retrospective study was performed by Christopherson et al. [69]. Patients undergoing surgery for colon cancer received a GA alone or GA with an epidural. The results showed the epidural group had a better survival only for the first two years.

Patients undergoing prostate cancer surgery receiving GA alone or GA with peridural anesthesia/analgesia were retrospectively compared. The results demonstrated that the epidural group had a better cancer recurrence-free of 57% compared to the control group.

The same type of anesthesia (with or without epidural) was used by de Oliveira et al. [70] to investigate retrospectively the recurrence-free interval in patients undergoing surgery for ovarian cancer. The results demonstrated that the addition of epidural was beneficial in terms of prolonging the recurrence-free interval.

Similar results were found in a retrospective analysis of patients receiving spinal anesthesia for excision of primary malignant melanoma of the lower extremity compared to the control group [71].

However, more recent retrospective studies did not show any benefits of adding a neuraxial block in terms of recurrence-free interval or mortality [70,71,72].

A recent meta-analysis [73] confirmed the discrepancy between the positive and negative results observed in the investigations dealing with neuraxial anesthesia and cancer outcome. Figure 1 shows the trial outcome from no effect to significant benefit versus year of publication and study size. It shows that the trends are first contrary to the first publications, more recent studies are rather negative and second, the larger study sample is associated with more negative results.

To summarize, it is well-known that the accuracy of medical records, the absence of randomization, blinding and standardization, and confounding factors are all serious bias, which can directly or indirectly influence the results and therefore do not allow for drawing valid conclusions.

## 7. Prospective Human Studies

Karmakar et al. [72] undertook a 5-year prospective follow-up of patients randomized to GA with or without thoracic paravertebral block undergoing a modified radical mastectomy. They found that the addition of a paravertebral block had little to no appreciable effect on local recurrence, metastases, or mortality. A similar study was conducted by Sessler et al. [74] In this trial, they found no difference in terms of breast cancer recurrence or mortality between those who received a paravertebral block or not; however, the major weakness of these two studies was the performance of only a single shot of LA in the paravertebral space, knowing that, after such a procedure, LA blood levels would be too low to modify the biology of the CTCs.

## 8. Conclusions

The perioperative period should be considered a crucial time in surgical oncology because chemotherapy will be delayed for a few weeks, opening a window for the occurrence of metastases. The application during this period of amide-type LAs may add a new option in the armentarium for the prevention of metastases.

In vitro and animal investigations have demonstrated the positive effect of lidocaine in the context of metastases on the biomolecular levels, including receptors and mechanisms.

Lidocaine, via different mechanisms, could inhibit/reduce cancer growth in vitro and in animal models. This provides valuable potential for its further application in cancer therapy and opens new insight for new drug discovery.

However, well-documented human studies in terms of dosage concentration and duration of time of application are needed to confirm the real benefit of this technique.

## Figures and Tables

**Figure 1 medicina-58-00882-f001:**
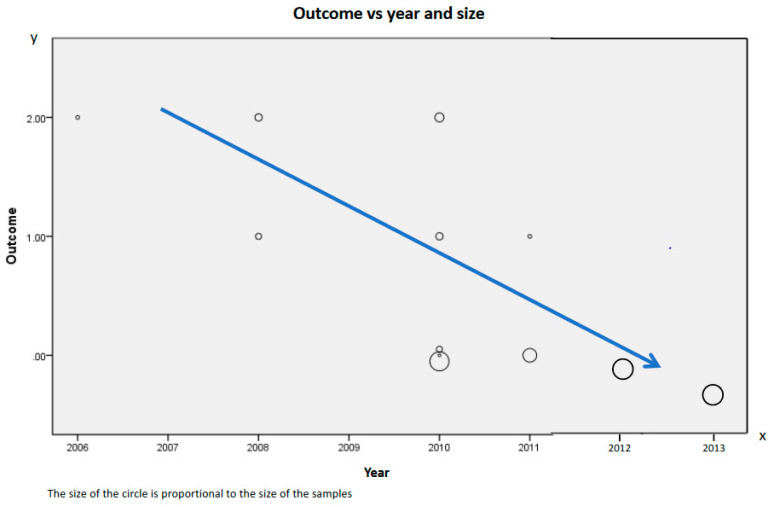
The outcome from 2 = positive result to 0 = null result is shown on the y axis. The year of publication is on the x axis.

**Table 1 medicina-58-00882-t001:** Natural killer cell activity: Influence of commonly used drugs in anesthesia.

	Human Studies	
	Without Surgery	With Surgery
Opioids		
Morphine	↑, =, ↓	=, ↓
Fentanyl	↑, =	=, ↓
Remifentanil	=	?
Ketamine	?	↑,↓
Propofol	?	↑
NSAID	↑	↑
Local anesthetics		
IV infusion	↑	↑
Regional application	(=)	(=)

↑ increase; = neutral effect; ↓ decrease; ? no available data.

**Table 2 medicina-58-00882-t002:** Biomolecular Mechanisms and Antimetastatic Properties of Amide-Type Local Anesthetics.

Inhibition of Src-Kinase/ICAM-1 Phosphorylation	Piegeler et al., 2014, 2012 [42,57]
Downregulation of VGSC (Voltage-Gated Sodium Channel)	House et al., 2010 [62]
Antiproliferative effects	Sakaguchi et al., 2006 [63]
Increase the apoptotic effect	Xing et al., 2017 [64]
Increase the demethylation	Lirk et al., 2012 [65]
Inhibition of cytoskeletal remodeling	D’Agostino et al., 2018 [66]
Potentiation in vitro and in vivo of NK cell activity	Ramirez et al., 2015 [32] Jaura et al., 2014 [33]

## Data Availability

Not applicable.

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
