# Peer review of "Impact of Local Anesthetics on Cancer Behavior and Outcome during the Perioperative Period: A Review"

_medicina, 2022, doi:10.3390/medicina58070882_

Round 1

Reviewer 1 Report

  1. The abbreviation in abstract should be explained, like “ICAM-1”.
  2. In the fourth paragraph of “Introduction”, the data sources in “... because more than 90 % of cancer mortality is caused by metastasis” should be explained and clarified.
  3. In the fifth paragraph of “Introduction”, the author discussed the influence of surgery on the development of cancer, which is not close-related to the topic of this paper, and the perspective should be further estimated.
  4. There have been many in-depth studies concerning CTCs, but the author only review the pancreas cancer extremely briefly. As a separated part, the content in this paper is not sufficient to support an entire part “The role of Circulating tumor cells” , which needs to be further enriched and extended.
  5. For part 3, NK cell is one of the various immune system cells, is there any other immune cells that influenced by local anesthetics? For part 5, are there any other local anesthetics exerting the effect on cancer cells? The part 3 and part 5 is confusing and may mislead readers. Additionally, it will be more systematic and clear to conclude the present evidence under different classification of either local anesthetics drugs or immune cells by table or figure.
  6. The abbreviations of “LPS” and in Part 4 and “GA” in Part 7need to be explained.
  7. Part 6 is too short to be a separate part, and though there may be not any positive results, it is professional and encouraged for a review article to conclude the present evidence in a more comprehensive, detailed and objective perspective.
  8. Editing of English language and style are required.

Author Response

  1. The abbreviation in abstract should be explained, like “ICAM-1”.
    We have explained the abbreviations

  2. In the fourth paragraph of “Introduction”, the data sources in “... because more than 90 % of cancer mortality is caused by metastasis” should be explained and clarified.
    This is a reality Patient do not die from the primary tumor, but from metastases.

  3. In the fifth paragraph of “Introduction”, the author discussed the influence of surgery on the development of cancer, which is not close-related to the topic of this paper, and the perspective should be further estimated.
    The title has been changed. Perioperative has been added.

  4. There have been many in-depth studies concerning CTCs, but the author only review the pancreas cancer extremely briefly. As a separated part, the content in this paper is not sufficient to support an entire part “The role of Circulating tumor cells” , which needs to be further enriched and extended.
    This para has been rewritten.

  5. For part 3, NK cell is one of the various immune system cells, is there any other immune cells that influenced by local anesthetics? For part 5, are there any other local anesthetics exerting the effect on cancer cells? The part 3 and part 5 is confusing and may mislead readers. Additionally, it will be more systematic and clear to conclude the present evidence under different classification of either local anesthetics drugs or immune cells by table or figure.
    Only NK has been studied in this context.
    The Part 3 has been rewritten.

  6. The abbreviations of “LPS” and in Part 4 and “GA” in Part 7need to be explained.
    The abbreviations have been explained.

  7. Part 6 is too short to be a separate part, and though there may be not any positive results, it is professional and encouraged for a review article to conclude the present evidence in a more comprehensive, detailed and objective perspective.
    This Para has been rewritten and completed.

  8. Editing of English language and style are required.
    The manuscript has been corrected by native English people.

Reviewer 2 Report

The covered topic is interesting and deserves to be considered by the scientific community. But the review, as proposed, is not acceptable for publication, because:

1) language problems:

a) Item 3. “Relation between local anesthetics anti-inflammatory properties, IMPROVED immune system Natural Killer (NK) cell activity and metastases

b) “NK cells, a lymphocytic subset of the innate immune system,17 designed to create the first defense against foreign substances” (A VERB IS MISSING)

c) The interaction between NK cell activity is dose dependent. (INTERACTION BETWEEN NK CELLS AND LOCAL ANESTHETICS?)

2) Structure and contents:

a) Item 3 says nothing about metastasis (as stated in the title).

b) Many definitions are not given in the text (eg CTC, LPS, GA,…)

c) Item 6 is too general and superficial. Please reformulate it! Moreover, such retrospective studies should be treated as Item 7, since they are also “human studies”

d) Item 7 “Human” studies are redundant

In conclusion

For me, as a review article, the effect of local anesthetics on metastasis prevention was not appropriated covered. The review seems superficial and uncommitted. I would suggest a deeper review on the literature.

Author Response

1) Language problems: The manuscript has been corrected by native English people

  1. a) Item 3. "Relation between local anesthetics anti-inflammatory properties, IMPROVED immune system Natural Killer (NK) cell activity and metastases:

Corrected, see new text.

  1. b) "NK cells, a lymphocytic subset of the innate immune system, 17 designed to create the first defense against foreign substances" (A VERB IS MISSING):

Corrected, see new text.

  1. c) The interaction between NK cell activity is dose dependent. (INTERACTION BETWEEN NK CELLS AND LOCAL ANESTETICS?):

Corrected, see new text.

2) Structure and contents:

  1. a) Item 3 sys nothing about metastasis (as stated in the title)

Corrected, see new text.

  1. b) Many definitions are not given in the text (eg CTC, LPS, GA, …..)

Definitions have been given.

  1. c) Item 6 is too general and superficial. Please reformulate it! Moreover, such retrospective studies should be treated as Item 7, since they are also "human studies"

Corrected, see new text.

  1. d) Item 7 "Human" studies are redundant

It is a prospective new study

In Conclusion: For me, as a review article, the effect of local anesthetics o metastases prevention was not appropriated covered. The review seems superficial and uncommitted. I would suggest a deeper review on the literature.

The text has been completed.

Round 2

Reviewer 1 Report

The authors have made some revisions of this manuscript. However, some critical problems still remain to be solved to make it a comprehensive and objective review.

1. Again, in the fourth paragraph of “Introduction”, the data sources in “... because more than 90 % of cancer mortality is caused by metastasis” should be explained and clarified by citing some reliable references.

2. Is there any other immune cells that influenced by local anesthetics?  For part 5, are there any other local anesthetics exerting the effect on cancer cells? Again, for part 3 and part 5, I strongly recommend the authors to conclude the present evidence under different classification of either local anesthetics drugs or immune cells by table or figure to make this review more systematic and comprehensive.

3. I strongly recommend the authors to organize present evidence reported from clinical trials by table in Part 6 to make it more clear and convinced.

Author Response

  1. Again, in the fourth paragraph of “Introduction”, the data sources in “... because more than 90 % of cancer mortality is caused by metastasis” should be explained and clarified by citing some reliable references.

Two references explained this point and have been included in the manuscript.

  1. Is there any other immune cells that influenced by local anesthetics?  For part 5, are there any other local anesthetics exerting the effect on cancer cells? Again, for part 3 and part 5, I strongly recommend the authors to conclude the present evidence under different classification of either local anesthetics drugs or immune cells by table or figure to make this review more systematic and comprehensive.

This para has been completed. A table has been included looking at the effects of usual anesthetics on the NK cell function. As explained all amide type anesthetics (lidocaine, ropivacaine, bupivacaine) have been shown to express these effects. On the contrary ester type LAs do not have any of these properties. A table summarizing the effects of amide-type LAs has been included in the manuscript.

  1. I strongly recommend the authors to organize present evidence reported from clinical trials by table in Part 6 to make it more clear and convinced.

Part 6: The text is more clearly explained and a figure has been added.

Round 3

Reviewer 1 Report

 Accept in present form.

This manuscript is a resubmission of an earlier submission. The following is a list of the peer review reports and author responses from that submission.